# Acceptability and perceived facilitators and barriers to the usability of biometric registration among infants and children in Manhiça district, Mozambique: A qualitative study

Olga Cambaco[1]*, Noni Gachuhi[2], Rebecca Distler[3], Carlos Cuinhane[4], Emily Parker[3], Estevão Mucavele[1], Quique Bassat[1,5,6,7,8], Célia Chaúque[1], Francisco Saute[1], Khátia Munguambe[1,9], Charfudin Sacoor[1]

1 Centro de Investigação em Saúde de Manhiça (CISM), Maputo, Mozambique, 2 The Global Good Fund I, LLC, Bellevue, WA, United States of America, 3 Element Inc., New York, NY, United States of America, 4 Faculty of Arts and Social Sciences, Department of Sociology, Eduardo Mondlane University, Maputo, Mozambique, 5 Barcelona Institute for Global Health (ISGlobal), Hospital Clínic-Universitat de Barcelona, Barcelona, Spain, 6 ICREA, Barcelona, Spain, 7 Pediatric Infectious Diseases Unit, Pediatrics Department, Hospital Sant Joan de Déu (University of Barcelona), Barcelona, Spain, 8 Consorcio de Investigación Biomédica en Red de Epidemiología y Salud Pública (CIBERESP), Madrid, Spain, 9 Faculty of Medicine, Eduardo Mondlane University, Maputo, Mozambique

* olga.cambaco@manhica.net

## Abstract

In low-and middle-income countries, many infants and children remain unregistered in both civil registration and healthcare records, limiting their access to essential rights-based services, including healthcare. A novel biometric registration prototype, applying a non-touch platform using smart phones and tablets to capture physical characteristics of infants and children for electronic registration, was tested in rural Mozambique. This study assessed acceptability and perceived barriers and facilitators to the usability of this biometric registration prototype in Manhiça district, southern Mozambique. The study followed a qualitative design consisting of 5 semi-structured interviews with healthcare providers, 7 focus group discussions with caregivers of infants aged between 0 and 5 years old, and 2 focus group discussions with data collectors involved in the implementation of the biometric registration pilot project. Data were thematically analysed. The results of this study show that there is wide acceptability of the biometric registration prototype among healthcare providers and caregivers. Participants were aware of the benefits of the biometric registration prototype. The perceived benefits included that the biometric registration prototype would solve the inefficiency of paper-based registration, and the perception of biometric registration as "healthcare norm". Perceived potential barriers to the implementation of the biometric registration prototype included: myths and taboos, lack of information, lack of time, lack of father's consent, and potential workload among healthcare providers. **In conclusion,** the biometric prototype was widely accepted due to its perceived usefulness. However, there is

**Data Availability Statement:** The data provided cannot be made available due to ethical constraints, as participants had not provided informed consent for their data to be stored in a public repository. Data can be requested at Centro de Investigação em Saúde de Manhiça's Internal Scientific Committee and Internal Ethical Review Board through the following addresses: secretariado.cibs@manhica.net– Telephone: (+258) 843163096, - CP1929 - Manhiça-Mozambique.

**Funding:** This study received support from The Global Good Fund I LLC, (USA; https://globalgoodfund.org/) in the form of salaries for authors NG, OC, CC, EM, and CC. Element, Inc. (https://www.discoverelement.com/) provided support in the form of salaries for RD and EP. The specific roles of these authors are articulated in the 'author contributions' section. The funders had no role in study design, data collection and analysis, decision to publish, or preparation of the manuscript. No additional external funding was received for this study.

**Competing interests:** The authors have read the journal's policy and have the following competing interests to declare: NG, OC, CC, EM, and CC are paid employees of The Global Good Fund I LLC, (USA; https://globalgoodfund.org/). RD and EP are paid employees of Element, Inc. (https://www.discoverelement.com/). This does not alter our adherence to PLOS ONE policies on sharing data and materials. There are no patents, products in development or marketed products associated with this research to declare.

a need to address the perceived barriers, and involvement of children's fathers and/or other relevant family members in the process of biometric registration.

## Introduction

In low-and middle-income countries, a considerable number of people remain unregistered in both formal civil registration and healthcare systems [1,2]. It is estimated that globally over one billion people are unregistered, with about half of these being children [1] and the majority living across Asia and Sub-Saharan Africa [3]. The Sub-Saharan African region accounts for over 250 million people without an official registration [1], and it is estimated that on the continent, only 1 out of 4 newborns is registered at birth [4].

Mozambique, a country in southern Africa, has approximately 12 million people without legal identity [1]; and less than half (48%) of children aged 0–5 years old are registered [5]. These statistics highlight that most infants and children remain unregistered, which may put infants and children at risk of misidentification [6]; lack of access to a birth certificate and official registration [7]; abandonments after discharge [8]; and subsequent challenges for accessing and benefiting from healthcare services, education, official identity and citizen's rights [9].

Several factors influence birth registration in sub-Saharan Africa. These include lack of awareness among parents and guardians about the importance of birth registration and certificates for their children future [10]; policies related to registration that do not often reflect cultural practices or religious beliefs; policies that impose fees or are punitive against delayed registration barriers to accessing registration centres, such as lack of transport or long distances; inefficiency of the current registration system used [11]; and lack of biometric system to register infants and children [12].

Moreover, in most sub-Saharan African countries like Mozambique, the most common registration system relay on paper-based methods [12]. Such methods have a number of disadvantages, including being time consuming, difficult to back-up, damageable or subject to loss during floods, fire, high humidity or emergencies, and not being user friendly in terms of returning the data to the central authority [11]. In Mozambique, when a child is born, healthcare providers issue a birth notification which is then used by the Civil Register Office (*Conservatória do Registo Civil*) to register the birth and issue a birth certificate. The registration is mandatory, and is free of charge when carried out within 120 days of birth [12].

Biometrics are measurable and distinctive anatomical physical characteristics or personal traits that are unique to an individual, which can be used for individual identification [13]. Physical characteristics of the person include face, fingerprint, iris, retina, hand [13,14], height, ear, body and hand shape [14]. Biometric data can also include behavioural features, such as signature, gait characteristics [13], body posture, speech, handwriting, heartbeat, eye blinking pattern [14].

A significant number of sub-Saharan African countries have started upgrading to biometric identity system for adults [15,16], especially for certain documents such as driver's licence, passports, or national identification [11,15]. However, in most countries, the biometric system does not yet include children [4]. Like other countries in the region, the use of biometric in Mozambique only applies to adults [11].

The use of a biometric identification system among infants and children is associated with a number of advantages. These include birth registration, access to unique identity for children [7], accurate identification and immunisation tracking of infants and children, access to universal health care, social protection, national ID and education [17]. Biometric identification

cannot be shared or misplaced [18]; it prevents duplication of registration and fraud [3]. Moreover, digital patient identification systems may improve health service delivery [19], or adequate and prompt follow-up of infants diagnosed or exposed to HIV infection [20], suffering from malnutrition [21], and other conditions requiring longitudinal tracking. It can also enhance patient privacy by preventing people from having to disclose their names and other identifiers in busy healthcare facilities [7,22].

At the same time, biometric systems have also been associated with risks such as coercion in the process of registration, exclusion when the system failures to identify, misuse of data when the database is stolen, or abusive use of the data [22], putting infants and children at risk [3].

In 2019, the Manhiça Health Research Centre (CISM) implemented a biometric registration pilot project in Manhiça district, a rural area in Southern Mozambique. The project tested a novel biometric registration prototype in its early stages of development in order to access the performance of the prototype and improve its robustness. This biometric registration prototype consists of a non-touch, mobile-based platform that uses smartphones and tablets. The implementation of the biometric registration project consisted of enrolment of caregivers who had children under 5 years old at health facility and community level. All the enrolled participants were informed about the potential importance of the biometric registration pilot project, its procedures and how the images would be stored before the starting of the project. A total of 8 field workers were recruited, trained and participated in the implementation of the project as data collectors. These data collectors used smartphones and tablets to capture physical characteristics such as the morphology of ears, feet and open palms of a total of 1920 newborns and children. No facial images of the newborns and children were captured. During the activities in the field, the images captured were immediately stored locally in the mobile and smartphones devices, and in the end of the daily activity, the devices were connected to the internet to upload the images to the CISM's server on a daily basis for on-site storage. Only the main researchers accessed the images.

This research therefore assessed acceptability, usability and perceived facilitators and barriers of the biometric registration prototype among healthcare providers, data collectors and caregivers whose infants and children were involved in the pilot biometric registration prototype in Manhiça district. The research also assessed acceptability and perceived barriers about biometric registration among caregivers whose infants were never enrolled in the pilot biometric registration prototype. This study was relevant because the successful implementation of the biometric registration may depend on the social and cultural values of the participant's population [23]. The biometric registration prototype implemented in Manhiça district represents a new method of registration, and therefore, it may challenge or influence the change in people's norms and practices–particularly if conflicting with local values and traditions.

The acceptability of a new experience like biometric registration prototype in the community, is mostly linked to the knowledge and experiences that the participant population has about the benefits of the system in their life or that of their infants and children. This assumption derives from Alfred Schutz's theory [24] of the life-world: Schutz's theory is related to people perceptions of ideas of the world presented to them, and it may enable us to understand how communities experience and perceive the biometric registration used to register infants and children. It also offers analytical tools to identify perceived facilitators and barriers regarding the implementation of biometric registration prototype in a specific social and cultural context.

Understanding how participant populations perceive and accept the biometric registration is essential because it could help in both the product design and the decision-making about whether the method should be further adopted and expanded across the country.

## Methods

### Study design and study sites

This qualitative study is part of a broader cross-sectional observational research in the study *biometric data collection in Mozambique infants and children*: *evaluation of an infant and child biometric prototype to accurately assess unique identity in southern Mozambique*. The qualitative research was conducted in the districts of Manhiça and Bilene-Macia, located in the southern region of Mozambique. Manhiça district, 80 km north of the capital Maputo, is located in the northwest of Maputo province, and spans to 2,373 square kilometres. It borders Magude district in the north, Gaza province in the northeast, Marracuene district in the south and Moamba district in the west [25]. Approximately 208,466 inhabitants lived in this district in 2017 [26]. There are 12 health centres and two hospitals, including the district hospital [26].

The Manhiça Health Research Centre (*Centro de Investigação em saúde da Manhiça*) (CISM), located in Manhiça district, has been running a demographic surveillance system in an area defined as Manhiça health and demographic surveillance site (Manhiça HDSS) since the year 1996 [27]. This research centre carries out among other activities, research on malaria, tuberculosis, diarrhoea, HIV and reproductive and maternal and child health [28], to improve population health through testing disease control interventions [27]. Some outcomes of the research findings produced by CISM have directly impacted the health of Manhiça's population both at local and national level [27,29].

The majority of Manhiça's population is rural, mostly engaged in small businesses or subsistence farming, or are labour in sugar cane plantations and sugar refining companies, and other small agriculture companies. The residents speak mainly Xichangana and Xitsonga. Some inhabitants also speak Portuguese, the official language nationwide. The predominant religion is Christian (dominated Zionists and Protestants) [25].

Bilene-Macia district has 2,157 square kilometres, and is located in Gaza province, south-western Mozambique. It borders with Chokwé district in the north, Xai-Xai district in the east, Magude district of Maputo province in the west and Indian Ocean in the south [25,30]. About 150,554 inhabitants lived in Bilene-Macia district in 2017 [26]. The district has 9 health centres [31]. The population of this district is mainly rural, and practice subsistence farming, small businesses, fishing, and some work in small agriculture companies and tourism industry [25,32]. The local inhabitants speak Xichangana, and some of them also speak Portuguese. The predominant religion is Christian related Zionism [25].

The two districts have similar characteristics. Both are rural and patriarchal communities. This means that an individual family's membership derives from and is recorded through the father's lineage. Inheritance of property, names, rights or titles passes through male kinship [33]. Concerning their social position in the household, men occupy the dominant position. The man is the head of the family and guardian of the children–while women occupy a subordinate position [34,35].

In Manhiça district, the qualitative study was conducted in both communities and health facilities, where the biometric registration prototype pilot project was implemented. These communities included Manhiça village, Maragra, Taninga, Palmeira, 3 de Fevereiro and Xinavane. The study was also conducted in Bilene-Macia district, particularly in Bilene-Macia village. However, Bilene-Macia district did not implement biometric registration pilot project. This district was included in the qualitative study for comparative analyse.

### Study participants, recruitment and data collection

The study participants were healthcare providers working in the health facilities where the biometric prototype was tested, caregivers whose infants and children participated in the pilot

biometric prototype system, caregivers whose infants did not participate in the pilot biometric prototype, and all study data collectors.

Recruitment and interviews took place between October 2019 and January 2020. The study applied purposive sampling to select both healthcare providers and caregivers; and it utilized semi-structured interviews and focus group discussions (FGDs) to collect data, as shown in Table 1.

All data collection tools were semi-structured, with mostly open-ended questions, organized in a logic sequence (from general to specific) allowing some, although limited, participant-driven expansion of the ideas being discussed.

Semi-structured interviews were applied to assess acceptability, usability and barriers of the biometric registration among the healthcare providers. A purposive method was used to select the healthcare providers, and the interviews took place in the respective healthcare facilities at selected times when they were available and had a lighter workload, and lasted between 16 and 40 minutes.

Focus group discussions were used to collect data with caregivers and data collectors to better assess the acceptability, usability and perceived barriers of the pilot biometric prototype from their perspective. In this study, caretakers were considered mothers or other adult female guardians aged 18 years and more, with infants or children aged between 0 and 5 years old. FGDs with caregivers comprised only women, and the size of each FGD varied between 6 and 10 members. A purposive method was used to access the members of FGDs among caregivers who participated in the pilot biometric prototype, while a convenience method was applied to recruit the members of the FGDs among caregivers who had not participated in the pilot biometric prototype.

FGDs with the same data collectors were conducted in two different periods. The first FGD with 8 participants was conducted during the biometric prototype study, and it assessed the feasibility of the platform used to register infants and children; while the second in the end evaluated the overall process of the biometric prototype registration system. FGDs with data collectors were 4 women and 4 men. FGDs during the study lasted between 80 and 120 minutes. The inclusion criteria for the participants were as it is presented in Table 2.

## Interview guide

Semi-structured interviews and FGD guides were developed to collect data with the study participants. The semi-structured guide for healthcare providers consisted of exploring the risks of the paper-based registration system used to register infants; feasibility, acceptability and usability of the biometric registration, as well as the perceived barriers for the implementation of the biometric system in the health facility. The FGD among caregivers explored the acceptability and the perceived facilitators and barriers about the biometric registration; while the FGD guide among data collectors focused on the evaluation of the biometric device, feasibility,

**Table 1. Data collection tools and sample size.**

| Data collection tools | Participants | Study sites and sampling | |
|---|---|---|---|
| | | **Manhiça** | **Bilene-Macia** |
| Semi-structured interviews | Healthcare providers | 5 | 0 |
| Focus group discussions (FGDs) | Caregivers with infants or children between 0 and 5 years old who participated in the pilot biometric prototype | 6 FGDs (n = 42) | 0 |
| | Caregivers with infants or children between 0 and 5 years old who did not participate in the pilot biometric prototype | 0 | 1 FGD (n = 10) |
| | Data collectors | 2 FGDs (n = 8) | 0 |

**Table 2. Inclusion criteria for the study participants.**

| Participants | Criteria for participation | Study sites |
|---|---|---|
| Health care providers | Age (over 18 years), Currently working in the selected health facility and familiar with the biometric prototype registration study and, Willingness to participate in the study. | Manhiça |
| Caregivers of children enrolled in study | Age (over 18 years) and having a child under 5 years old who participated in the biometric testing and, Willingness to participate in the study. | Manhiça |
| Caregivers of children NOT enrolled in study | Age (over 18 years) and having a child under 5 years old, Having no child recruited and participated in the biometric registration testing; and Willingness to participate in the study | Bilene-Macia |
| Data collectors | Being data collector who participated in the process of the biometric prototype registration system data collection. | Manhiça |

their experiences with caregivers during biometric registration and the perceived barriers regarding biometric registration. The guide of semi-structured interviews a FGDs were designed according to the research objectives focusing the main relevant elements of the piloted biometric prototype. Each guide focused on specific topics, but remained opened to the emergence of new related themes relevant to the study object.

## Procedures

The study obtained ethical clearance from CISM's Internal Scientific Committee, protocol number Ref: CC/034/SEPT/2008 and the Internal Ethical Review Board, protocol number Ref: CIBS-CISM/058/2008. Verbal information about the objective of the study was provided. Written, informed consent was obtained from all participants. Interviews and FGDs were conducted on the language the participants found most comfortable. All healthcare providers were interviewed in Portuguese while all FGDs with caregivers were conducted in local languages. All interviews were audio recorded following the consent of the participants. Three Social Scientists researcher of CISM collected data: two female and one male researchers. The researchers were under a supervision of a female research coordinator. All researchers conducted semi-structured interviews and FGDs. Each FGD was conducted by two researchers: one played a role of moderator and another recorded and took notes of the non-verbal behaviours and dynamic of the discussion. All researchers, including the research coordinator listened and evaluated each semi-structured interview and FGD before the performance of other interviews and FGDs. This process enabled to ensure the quality of issues discussed, identify and address possible gaps during data collection.

## Data analysis

All audio recorded data were independently transcribed. A total of four researchers: three researchers who collected the data and the research coordinator controlled the quality and accuracy of the transcriptions, comparing the audio recording with the written transcriptions and correcting them when was necessary. All approved transcriptions were shared with all members of the research team, who then read and preliminarily coded the interviews. The research team discussed and decided on the preliminary cods and categories emerging from the data, and NVivo software, a qualitative package for qualitative data analysis, was employed to summarize the data. A content thematic analysis approach [36] was used to define the themes emerging from the data. The identified themes and subthemes were discussed, refined and revised by all members of the research team. The subthemes enabled to identify relevant content in participants' interviews, which were used to support each theme. The generated

final themes were: perceptions of healthcare providers regarding the actual registration system used to register and identify children; acceptability of biometric registration prototype, and perceived facilitators and barriers of usability of the biometric registration prototype. These themes are presented in the results section.

## Results

### Sociodemographic characteristics of the study participants

The participants of this study comprised healthcare providers, caregivers with or without biometric experience and data collectors. Among healthcare providers, 4 participants were female and 1 was male, all had specialised trained in primary healthcare and had more than one year in their performing duties (Table 3).

The majority of caregivers of Manhiça and Bilene-Macia districts were aged between 18 and 55 years old, married or living with a partner, and were housewives. The majority of the participants had some formal education as shown in Table 4.

The study also included FGD with data collectors with the following characteristics: 4 were male and 4 were female, 3 data collectors had completed secondary school, while 1 had bachelor (Table 5).

### Perceptions of healthcare providers regarding the system used to register and identify children at the healthcare facility

**Healthcare providers' accounts of registration and registration process at the healthcare facility.** Newborn registration at the healthcare facility occurs immediately after childbirth and is performed by the healthcare providers. All healthcare providers interviewed reported using a paper-based system to register newborns in a 'maternity registration book', including baby's weight, birth date, place of birth, sex, mother's name, address and contact information.

One of the healthcare providers stated:

"*We register the baby as soon as he or she is born. We use a maternity book, where we record baby mother's name, weight, sex of the baby, birth date, and the address and contact of the*

**Table 3. Sociodemographic characteristics of healthcare providers.**

| Characteristics of participants | Frequency |
|---|---|
| **Sex** | |
| Male | 1 |
| Female | 4 |
| **Age range** | |
| 25–35 | 2 |
| 36–40 | 3 |
| **Educational level** | |
| Secondary school | 5 |
| **Specialisation** | |
| Maternal health nurse | 3 |
| Preventive medicine | 2 |
| **Working experience** | |
| 1–2 | 1 |
| 3–4 | 2 |
| 5 and more years | 2 |

**Table 4. Sociodemographic characteristics of caregivers.**

| Characteristics of participants | Manhiça (n = 42) n/% | Bilene-Macia (n = 10) n/% | Total (n = 52) n/% |
|---|---|---|---|
| **Age range** | | | |
| 18–25 | 26 (69,9) | 5 (50) | 31 (59,6) |
| 26–35 | 12 (28,6) | 5 (50) | 17 (32,7) |
| 36–46 | 4 (9,5) | 0 | 4 (7,7) |
| **Educational level** | | | |
| None | 3 (9,5) | 1 (10) | 4 (7,7) |
| Uncompleted primary school | 11 (26,2) | 2 (20) | 13 (25) |
| Completed Primary school | 8 (19) | 2 (20) | 10 (19,2) |
| Uncompleted secondary school | 17 (40.5) | 4 (40) | 21 (40,4) |
| Completed secondary school | 3 (7,1) | 1 (10) | 4 (7,7) |
| **Marital status** | | | |
| Single | 15 (35,7) | 1 (10) | 16 (30,8) |
| Married/living with a partner | 27 (64,3) | 9 (90) | 36 (69,2) |
| **Occupation** | | | |
| Housewife | 41 (97,6) | 10 (100) | 51 (98,1) |
| Factory worker | 1 (2.3) | 0 | 1 (1,9) |
| **Religion** | | | |
| Zion Christian Church | 27 (64,3) | 6 (60) | 33 (63,4) |
| Assembly of God | 7 (16,7) | 4 (40) | 11(21,1) |
| Christian Catholic church | 1 (2,3) | 0 | 1 (1,9) |
| Nazarene Church | 2 (4,8) | 0 | 2 (3,8) |
| Apostolic Church | 2 (4,8) | 0 | 2 (3,8) |
| Pentecostal church | 3 (7,1) | 0 | 3 (5,7) |

*mother. We do not immediately register the baby's name because the mothers do not mostly give their babies' names soon after birth. They normally give names to their babies at their homes after discharge from the healthcare facility. We only have access to the baby's name two months after childbirth, and then we add the baby's name in the maternity book"* (Healthcare provider, Palmeira healthcare facility).

**Table 5. Sociodemographic characteristics of data collectors.**

| Characteristics of participants | Frequency |
|---|---|
| **Sex** | |
| Male | 4 |
| Female | 4 |
| **Age range** | |
| 24–29 | 4 |
| 30–35 | 4 |
| **Educational level** | |
| Secondary school | 7 |
| Graduated (Bachelor) | 1 |
| **Marital status** | |
| Single | 4 |
| Married/living with a partner | 4 |
| **Religion** | |
| Christian Catholic church | 7 |
| Apostolic Church | 1 |

All healthcare providers also said that they use a manual system to identify babies. This system consists in assigning a small colour-coded card or paper (pink for girls, blue for boys) with the mother's name, which is tied into the baby's wrist immediately after birth. According to the healthcare providers this method is also used to prevent the exchange of the babies; as one of the participants explained.

"*When the baby is born, we write the mother's name in a small rose or blue card, and we put it in the wrist of the baby; that is the norm. But sometimes, these coloured cards run out; and we find out other alternatives, such as cutting a small paper and writing the name of the mother's baby. This method helps us to identify the baby and also to prevent that the baby is not exchanged*". (Healthcare provider, Xinavane Healthcare centre).

According to the healthcare providers, before the mother and the baby are discharged, the baby receives a personal identification and health record's card. This card, known as the child health card or "yellow card" due to its appearance (*Cartão de saúde da criança or cartão amarelo*), is later used to identify the child at all subsequent healthcare consultations. This card contains all vital newborn data such as parents' names; address and contact of the mother, birth date, weight and book registration number, and is used to record any relevant clinical data during health care visits, including weight curves and vaccinations received.

**Healthcare providers' perceptions about the paper-based registration system.** All healthcare providers classified the paper-based registration as manual. Some healthcare providers said that the method was not feasible because any mistake during registration process would affect correct identification of the baby. One of the healthcare providers explained it as follows:

"*The great problem of this manual method is related to mistakes in or the absence of the registration of the baby or mother of the baby. Sometimes healthcare providers make mistakes in the process of registration, and when it happens, the identification process is no longer possible. It is necessary to go back and check the maternity book in order to certify if the mother gave birth in that healthcare centre, and if she is really the mother of the baby.*" (Healthcare provider, Taninga healthcare facility).

Other healthcare providers viewed the paper-based registration as problematic because it does not enable a quick identification of the infants and children. They stated that it usually takes a long time to identify the baby during consultation, especially when the mothers forget or lose the child health card; or someone else–usually other members of the family–takes the baby to the hospital without a child health card. One of the healthcare providers expressed her views as follows:

"*The problem of the actual system is that when the mother forgets or loses the child health card, we have to check the data in the registration book. This takes a long time, and most often you do not find the name of the child or the book. Sometimes the child is transferred to another healthcare sector, and the book is not there, and there is no way the child can be identified without a child health card. When it happens, we often try to use the physical characteristics of the child to define the approximate age. This is often a risk we have to deal with because the child physical characteristics do not often lead to the accurate age. This is because most often mothers or others caregivers such as grandmothers do not know the age of the baby, but we have to take care of the child, we have no other way; we cannot send the child back home without treatment.*" (Healthcare provider, Maragra health facility).

Moreover, some healthcare providers said that paper-based registration run the risk of double registration. This often occurs when mothers go to the health consultation without the child health card, and the child is not identified in the maternity book. When this happens, the healthcare providers register the child again. Healthcare providers also said that the paper-based registration system was not robust enough to keep child data for long periods of time because after 3 or 4 years, the book could disappear or some pages could be lost, leading to the loss of all relevant data of the children; as mentioned by one of the healthcare providers:

"*The child data are recorded in the maternity book. However, after 4 years and some months, we cannot go back to the maternity book because it is difficult, especially when the mother does not even remember the child's birthdate and the registration date (. . .). Our books do not help because sometimes, some pages are lost. . .they are books, we all hold the same books, and after 2, 4 years it is not easy to find the books in a good condition*". (Healthcare provider, Palmeira healthcare facility).

Likewise, all healthcare providers said that the child health card was not a solid method for the identification of the newborns and children because most mothers often lose it during their travels, flood events or theft. They also said that most mothers do not often remember all relevant data of the children, such as child birthdate, registration number, and type of consultation previously received. Moreover, the healthcare providers reported that in most communities of Manhiça district caregivers of the children were not only the infant mothers, but also grandmothers or other family members, who can also take the baby to the health facility, disrupting in this way the continuum of information tracing. For instance, caretakers, such as grandmother do not often know the name and age of the baby. According to the healthcare provider, all these problems lead to misidentification of the infants and children.

Misidentification also occurs when the child health card presents different data from that provided by the mother. Some healthcare providers said that when this happens, they often call to the healthcare facility where the mother gave birth to find out what happened and try to correct the data. Moreover, other healthcare providers also reported informing the local community leaders to sensitize the population to avoid requesting someone else to take the baby to the healthcare facility without the child health card. Additionally, healthcare providers said that they often request the community leaders to locate the child and the mother when they miss appointments at the health facility, especially when a mother gave a wrong address or contact.

**Perceived impact of misidentification of the infants and children at the healthcare facility.** Healthcare providers said that misidentification of infants and children leads to several risks, such as inability to access to accurate healthcare service, and thus health risks to infant and children, but also risks to the health sector. Individual risks may occur, for example, when a medication is administered to a child without knowing his or her accurate age. This often happens when caregivers forget or lose the child health card and they cannot recall the exact age of the baby. One of the healthcare providers explained it as follows.

"*The procedure is that all healthcare service practices are associated to the age of the baby. For example, it is the age that determines where the child must receive the treatment. When the child is 2 months old, he or she must receive treatment at postnatal service, but when the child has 2 years, the place to go is the paediatric service. Moreover, the type and quantity of the medication to be administered is also determined by the age. So, when the mothers do not know the correct age of the child, then the quality of service delivered is affected*". (Healthcare provider, Xinavane healthcare facility).

Some healthcare providers also perceived that lack of identification may lead to overdose or intoxication of the children, causing serious risks to the health of the child. One of the healthcare providers expressed this concern as follows:

"*Misidentification may lead to several problems: the child can be prescribed wrong medication or overdose because we do not know the accurate age. Moreover, some children receive supplements, and when the mother loses the child health card, we do not know which supplement must be prescribed. For example, without accurate information, we do not know if the child has already received vitamin A, and we can decide to prescribe vitamin A again, while the child has already received it. When there is a lack of child information we estimate the age of the baby, and we prescribe the medication. But this is dangerous as it may lead to intoxication of the child and increase child health problems because inaccurate medication can lead to disease resistance.*" (Healthcare provider, Palmeira healthcare facility).

Moreover, healthcare providers perceived that misidentification of infants and children may lead to risk to the health sector. Some healthcare providers said that misidentification negatively affected the health data quality produced at the healthcare facility. They also added that inaccurate patient data leads to problems of medical prescription; while others said that misidentification led to the default of the medical rules and, they perceived that when they prescribed without accurate age, the prescription was similar to home prescription, which does not follow medical rules. One of the healthcare providers explained it as follows:

"*When the prescription is based on the estimation of the age, not the accurate age, then this is similar to home prescription because we are not following the medical rules, the prescribed medication is not based on the age and weight of the child*". (Healthcare provider, Xinavane healthcare facility).

Some healthcare providers also felt guilty when they prescribed medication to children without accurate information, and they perceived that the healthcare sector was lacking its responsibility to address accurately the problem of the patients. They added that anything that could happen to the child would be healthcare provider's responsibility as well as health sector responsibility.

## Acceptability of using biometric registration for infant and children registration

**Acceptability of using biometric registration among healthcare providers.** Biometric registration was welcomed and widely accepted among healthcare providers. Many perceived advantages contributed to this acceptability. The biometric registration was regarded as useful and helpful for infant and child registration, identification and follow-up at the healthcare facility; as one of the healthcare providers stated:

"*With biometric registration, it would be easy to register and identify infants at the healthcare facility because even if the child and the mother have already left the maternity ward, we would already have had all relevant data in the system. Whenever the child comes back, we would access his or her data in the system and do the follow-up*". (Healthcare provider, Palmeira healthcare facility).

Moreover, some healthcare providers said that the biometric registration would help them to manage child health service, and it would bring positive contributions to access to

healthcare service, quality of data and identification, quickly and accurate access to child health data, access to accurate child identity during consultations and in the future sickness events, as well as enabling the identification of lost or stolen children and prevent baby exchange. Other healthcare providers perceived that the biometric registration would bring solutions to the actual challenge related to identification of infants and children, especially those whose mothers forget or lose the child health card; and allocate children to appropriate child health service. Healthcare providers also perceived that the biometric registration would positively contribute to accurate diagnosis and prescription of the recommended medication without mistakes. One of the healthcare providers explained it as follows:

> "*The biometric registration will help to diagnose because all the information such as age, birth day, place of birth, health related problem, would be already in the system; and that will facilitate us to prescribe the correct medication*" (Healthcare provider, Maragra healthcare facility).

Furthermore, healthcare providers said that the biometric registration would reduce the time actually spent identifying infants and children. They felt confident about possible positive changes with the implementation of biometric registration in the healthcare facilities. One of the healthcare providers presented the following view:

> "*Our actual registration system* [paper-based registration] *is time consuming. With the introduction of biometric registration, it will take short time to identify the child; I mean in 5 or 10 minutes we will be able to identify the child. So, I think the biometric registration is a good thing, it will bring positive changes*". (Healthcare provider, Maragra healthcare facility).

**Acceptability of using biometric registration for infant and children among caregivers.** The biometric registration was also widely accepted among both caregivers with or without experience of biometric registration. Caregivers of all communities said that they accepted or would accept to participate in the biometric registration pilot project because i) the biometric registration would benefit their children, ii) it would enable identification of children at the healthcare facility, and iii) they perceived that it was mandatory to participate because it was healthcare facility norm.

Both caregivers with or without experience of biometric registration involved in this study were in consensus that the biometric registration was very important to their communities because it would benefit primarily their children. For caregivers with experience of biometric registration, children who participated in the pilot project would benefit from early diagnosis of possible childhood diseases and good care at the healthcare facility. Similarly, some caregivers without experience of biometric registration said that they wanted the biometric registration because it would improve their children's health. Moreover, participants said that they agreed or would accept to participate in the biometric registration project because the registration would enable the identification and health treatment of the child at the healthcare facility even without the child health card. Some participants said:

> "*I accepted my child to be photographed because as they explained to me, in case I travel to South Africa or somewhere else, and I lose or forget the child health card, the child will still be identified and treated at the healthcare facility. Based on this explanation, I did not have any doubt about the benefit of the project*". (FGD with caregivers, mother, participant 1, Maragra healthcare facility, Manhiça district).

"*It is important that children are photographed because nowadays when the child is born at the healthcare facility, she or he can be stolen. But if the child is photographed, and the palm-print of the mother is saved, if the baby is stolen, then she or he will be found somewhere else, and this will be easy to identify based on the photographs. That is why it is very important to photograph the babies*". (FGD with caregivers without experience of biometric registration, participant 1, Bilene Macia distric).

Some caregivers with experience of biometric registration reported accepting to participate in the registration pilot project because they perceived it as mandatory, as a norm of the healthcare facility. Other caregivers without experience of biometric registration also perceived the possible introduction of biometric registration as "*healthcare facility norm*" or "*government rule*", which should not be denied. Both participants perceived that anything from the healthcare facility was good and better for their child's health, and it was mandatory because the healthcare facility was the main place responsible for their children's healthcare. This is how one of the participants expressed her feeling:

"*We cannot deny because it is healthcare facility norm. When they find you and request to examine or do something to the child, you cannot deny as long as they come from the healthcare facility. There is nothing you can deny under healthcare facility norm. The healthcare facility norm obliges us to live with this. We all have to accept; we accept our children to be photographed*". (FGD with caregivers, mother, participant 1, Taninga healthcare facility, Manhiça district).

The perception of participating in the registration pilot project as healthcare facility norm was also extensively perceived to most caregivers and communities. The participants with experience of biometric registration revealed that most other caregivers of their neighbourhood also viewed the participation in the registration as healthcare facility norm.

Healthcare providers also reported high acceptability of mothers in the biometric registration at the healthcare facility. They said that all mothers who were approached and invited to participate in the biometric registration accepted to participate, and all their children were photographed. Similarly, data collectors experience also shows that there was wide acceptability of the biometric registration in the community. Most of data collectors reported that they were well received in the selected families. They also said that some caregivers' partners regarded the biometric registration as useful to their children, as one of the participants presented the fieldwork experience.

"*Yes, the majority of families accepted and welcome the biometric registration. I think fathers were also happy to have their children photographed because when they were explained that that was a healthcare facility gadget, they said it will be useful to prevent children to be stolen. They said there are thieves who steal our children, and if the child is stolen, then the healthcare facility would easily recognise our children, and we will have them back*". (FGD with Data collectors, Manhiça district).

**Acceptability regarding the infant age and parts of the body for photographing.** All healthcare providers perceived that the infants should be photographed as soon as they are born to enable access to relevant child data. Also, both caregivers with or without experience of biometric registration said that the infants could be photographed immediately when they are born. One of the caregivers expressed her opinion as follows:

"*The infant can be photographed at any age. We accept. We cannot deny that. We, mothers, start receiving treatment during pregnancy for the better of our babies. Healthcare providers diagnose and inform us about the health of the baby, they start treating the baby before he/she is born. It is similar to photographing; it is for the benefit of the child. Children can be photographed even immediately after birth*". (FGD with caregivers with experience of biometric registration, participant 1, Taninga healthcare facility, Manhiça district).

Both caregivers with or without experience of biometric registration accepted to photograph any part of the body of the infant and children. They perceived that healthcare providers determined the parts to be photographed and they could not deny because they did not know what was the better for their children. However, some caregivers without experience of biometric registration said that they would not accept photographing certain parts of the child body such as ear because they could not see the usefulness of such an image.

Caregivers with experience of biometric registration reported that the process of photographing was not difficult, and they were not afraid because they received an explanation about the process. Moreover, most caregivers whose children participated in the biometric registration said that the time spent during photographing process was acceptable, as it was not too much time. However, some caregivers reported the inconvenience of the lengthy photographing process, which occurred only as a requirement of piloting and validation process. The timing issue stopped from continuing with their activities until the photos were completed.

## Perceived facilitators and barriers regarding the usability of smartphones and tablets for biometric registration

**Perceived facilitators regarding the usability of smartphones and tablets for biometric registration.** All healthcare providers welcomed the use of smartphones and tablets for registration of the infants and children. They also said that there were enough human resources at the healthcare facility, such as maternal child health nurses and other healthcare providers who are responsible of child healthcare, who if are trained could use the biometric registration to register and identify infants and children. Moreover, some healthcare providers were familiar with tablets and smartphones similar to those used in the biometric registration to record data in previous several studies conducted in the healthcare facility where they work, and they viewed these devices as of easy management. One of the healthcare providers explained it as follows:

"*I use a similar device to record data, photos and other information during different studies here* [healthcare facility]. *For example, I have a tablet, which I use to record data such as weekly epidemiological bulletin, and I directly send it to the Ministry of Health. It is easy to use a gadget like this tablet because it is an instrument that we already know how to use it. Apart from the health service, we also use both tablets and mobile phones in our personal issues. It is very easy to use these devices*". (Healthcare provider, Maragra, healthcare provider).

Additionally, healthcare providers had a good impression about the digital device used in biometric registration in terms of its usefulness and possible benefits in their everyday work. They perceived that the device would bring a positive change in their work routine. The benefits and changes mentioned include improving the quality of work performed, better organization of the infant data collected, and saving time in registration and identification of the infants and children.

Both caregivers with or without experience of biometric registration regarded smartphones and tablets used for biometric registration as useful innovation, which would benefit both children and mothers. Furthermore, the participants trusted data collectors because they were viewed as health workers. This perception favoured the implementation of the biometric registration pilot project particularly among caregivers who participated in the biometric registration.

Data collectors also confirmed that most caregivers adhered to the biometric registration pilot project because they perceived it as part of the healthcare facility and CISM activities, two institutions with which they had already built rapport and trust. According to data collectors, CISM and its personnel who were involved in biometric registration were often associated to the local healthcare facilities, and therefore, people were motivated to participate in the project.

Access to information about the biometric registration was viewed as a very import way to mobilize people to adhere to biometric registration. All participants, healthcare providers, caregivers with or without experience of biometric registration and data collectors, reported that caregivers were more prone to adhere to biometric registration when they were correctly explained about how the registration worked and its benefits. One of the caregivers with experience of biometry system confirmed it as follows:

"*When those who photograph explain to you about the benefits of it and the way the system works, there is no way you can deny it. Even if the process takes time, you will be patient because you know the correct benefits of the registration*". (FGD with caregivers with experience of biometric registration, participant 10, Palmeira, Manhiça district).

**Perceived barriers about the usability of smartphones and tablets for biometric registration.** All participants, healthcare providers, caregiver with or without experience of biometric registration and data collectors, perceived several barriers about the usability of the biometric registration. Barriers among healthcare providers included workload in their routine service, possible problems related to the functioning and management of the database, time spent to register infants, community myths and taboos.

Some healthcare providers reported that biometric registration would potentially increase the amount of their routine work, bringing negative impact in their routine service, if its usability had to be combined with the paper-based system already used. Others were concerned about how the biometric registration would work. They said that sometimes they faced problems related to lack electrical power and access to internet; and this would hinder access to digital database for infant identification. They also perceived that the biometric registration, particularly collecting the infant images, was time consuming, and that could discourage caregivers to adhered to biometric registration. Moreover, some healthcare providers reported that some community myths and taboos could hamper caregivers to adhere to the digital registration. These included community perception of possible use of infant images for witchcraft or child theft; as one of the healthcare providers explained it.

"*We have many myths and taboos in the communities. People often think and associate any activity to anything. for example, when we introduced altimeter for child weight and height, they associated it with the measurement of the coffin of the infant. So, they may think that the child images collected are for witchcraft. Most people often think and associate many things to witchcraft. They may also think that the child images are for child theft, because there have been many cases of child robbery here*". (Healthcare provider, Taninga healthcare facility).

Apart from these myths and taboos, healthcare providers also said that caregivers may also be afraid about how the child images would be used. They said that caregivers might think that the pictures could be sold or published in social networks, such as Facebook and WhatsApp. One of the healthcare providers explained it as follows:

"*People may not adhere to biometric registration due to fear. They may fear the child images will be sold or published in Facebook or WhatsApp. People often get worried whenever I use my mobile phone in front of then, they wonder if I am taking their picture or recording; thus, they may raise questions about what their children images are for and where will they be loaded or published*". (Healthcare provider, Xinavane healthcare facility).

All participants, healthcare providers, caregivers and data collectors alike, also mentioned that lack of consent of the child's father could hinder the biometric registration. They said that though most caregivers accepted medical norm, they might need their husbands' consent to photograph the infants and children. They perceived that adherence of the female caregivers will be dependent on children fathers' consent. For example, data collectors also faced lack of the child's father consent in some families; as one of the participants narrated his experience.

"*There were other families where we could not start photographing the children because the caregiver said that we have to talk to the child father. They could not allow us before this procedure. But when we talked to the child father he did not accept to participate*". (FGD with data collectors, participant 2, Manhiça distric).

Moreover, family members, particularly grandmothers seem to play an important role in caregivers' decision making about child healthcare. Some caregivers with experience of biometric registration said that other potential caregivers did not participate in the biometric registration because their mothers-in-lows did not allow it. One of the participants said:

"*My neighbour did not accept it. After photographing my children, we went to her house. But who did not accept was not* her [the child's mother], *it is her mother-in-law. She said: "my grandson cannot be photographed because I do not know what this is for". Even after the data collector had explained her the usefulness of the child images, the mother-in-law said she did not want it because even herself got children and they were not photographed while they were young*". (FGD with caregivers with experience of biometric registration, participant 2, Maragra, Manhiça district).

All participants, healthcare providers, caregivers and data collectors, also said that lack of information could be the main barrier of usability of biometric registration. They stated that without information caregivers will not adhere to the registration. For example, some participants with experience of biometric registration said that some caregivers in the community who were approached did not adhere to biometric registration because probably they lacked access to proper information. However, all participants believed that most of the earlier mentioned barriers would be overcome if people received appropriate information about the biometric registration.

Caregivers, in particular, also mentioned a number of possible barriers which could hinder the usability of biometric registration in their communities. The perceived barriers included fear associated to the robbery of children, perception of registration as time consuming, unfitness of the new registration method to common registration experience. Some caregivers with experience of biometric registration said that they were afraid to have their children

photographed because they were concerned about the usefulness of the images of the child, and they thought the images of the children would be used for stealing the children. Other caregivers were afraid because their infants were still very young, and they viewed the photographing process as punishment of the infants because they considered the newborns as sacred; while others thought that something bad could happen to the infant, such as getting ill or even die after photographing. One of the caregivers shared her feelings as follows:

> "*I was afraid because I thought the baby could die during photographing process or it is a way to take my baby from me, you never know, everything can happen*". (FGD with caregivers with experience of biometric registration, participant 5, Maragra, Manhiça district).

Caregivers with experience of biometric registration also said that photographing did take up too much time. According to these caregivers, most caregivers reported lack of time and they perceived the photographing as time consuming. Likewise, some participants said that other caregivers did not participate in the pilot project because the biometric registration was new and they perceived it as strange. They added that most caregivers said that they had been bearing children for long time and registering them, but they had never heard that children had to be photographed.

Data collectors who participated in the biometric prototype registration pilot project reported facing several barriers during image collecting in the communities. These barriers included lack of identification of the mothers recruited at the healthcare facility during follow-up, perceived stress of the mother during photography process. Some data collectors said that after recruiting caregivers at the healthcare facility they could not find them in the community during follow-up because some caregivers often changed their address or travelled to somewhere else. Additionally, they reported that some caregivers felt uncomfortable during child image collection and they pulled out. Moreover, they perceived that some caregivers felt stressed because of the length of time it took to capture all the required images for the prototype device.

Regarding the device used to photograph children; data collectors perceived that smart phones were better than tablets because they were easier to handle. Tablets were perceived difficult to hold while taking images at the same time. Moreover, some data collectors perceived that it was difficult to take images of the palm of a newborn, especially in the follow-up because they were difficult to match with the previous image.

## Discussion

To the knowledge of the authors, this is the first qualitative study conducted in Mozambique analysing acceptability and perceived facilitators and barriers about the usability of a biometric registration prototype for infants and children. The results of this study suggest that both healthcare providers and caregivers with or without experience of biometric registration widely accepted to use the biometric registration prototype to register infants and children. They also agreed that infants could be immediately registered after childbirth. In particular, both caregivers with or without experience of biometric registration accepted that any part of the body of the child could be photographed and saved in the biometric registration database of the healthcare facility.

Acceptability varied according to the perceived usefulness of the biometric registration amongst the study groups. While the healthcare providers perceived that the biometric registration would be useful to register, identify and follow-up infants and children, enhance adequate access to healthcare service, diagnosis and prescriptions; both caregivers with or without

experience of biometry registration viewed the biometric registration as useful to their children and themselves because it would improve their children's general health through correct identification of children at the healthcare facility and also prevent the swap and theft of children. These results suggest that the acceptability of biometric registration among healthcare providers and caregivers is associated to their understanding of its benefits. Such knowledge about the benefits of the biometric registration system was acquired through contact with the data collectors during the implementation of the biometric registration pilot project. As Schutz [24] stated, people often use their knowledge to interpret and evaluate events of their communities. In this regard the acceptance of the biometric registration by both healthcare providers and caregivers derives from what they learned about it, and as a result of frustrations with the inefficiencies of the current paper-based registration system. Like Storisteanu et al. [22] noted, the paper-based system is fragile, resource intensive, and it may fail to identify and follow-up patients, while the biometric registration not only solves all the related problems of paper-based identification, but it can also improve patients' health access and save lives.

Indeed, there are several benefits of biometric registration in the health sector [37]. The implementation of fingerprint to identify adult patients has shown promising results in patients' health improvement. In Zambia, for example, the implementation of fingerprints to identify female sex workers was found feasible and accepted, and the system improved healthcare service delivery in this group [38]. In general, the use of fingerprint in health sector is associated to high accuracy and secure identification, privacy of the patients, enables service to be delivered to the intended beneficiaries, vaccine coverage, support civil registration and vital statistics system [22,39]. Moreover, some pilot projects regarding the use of mobile devices for birth registration have also shown good feasibility in rural areas of Uganda and Senegal [40]. In Senegal, for example, mobile phones equipped with specific software were distributed to villages chiefs, who then captured the information regarding birth in their villages and transferred that information to the State Registrar for electronic registration; while in Uganda the hospitals were equipped with a specific program to access a web-based application to register birth, and villages chiefs were given SIM cards mapped to their names to register births locally and send the information to the local hospitals [40].

Despite receptiveness of the biometric registration among healthcare providers and caregivers, all participants were aware of the challenges of the new biometric prototype registration system. The perceived barriers presented by all participants of this study are consistent with the concerns of the use of biometric system in a general population, and with infants and children in particular. Studies [7,23,41–44] had already revealed that biometric system poses risks with relation to data reliability, reusability, security and its social impact [44]. Moreover, many people fear the misuse or exploitation of personal data for other unknown or inadequate purposes [23]. The use of biometric system in the health sector, in particular, has generated concerns about privacy and security of their data [38,42]. The use of biometric systems among infants and children has also been associated with risks, such as misuse or authoritative use of child data, leading to violation of privacy of children and their families [22,45,46], or exclusion of some children due to difficulties in capturing their biometric traits [3]. As it related to this, technological risk, data collectors interviewed in this study confirmed that it was difficult to capture and match the palm of the young infants. If not corrected in product design, this challenge may contribute to future difficulty in identifying these children, and therefore leading to exclusion in access to the health service.

Moreover, infants and children can be excluded from the biometric registration when their fathers or other relevant family members, such as mothers-in-law do not consent it. This is an important cultural factor that has to be considered because in most regions of Mozambique, in general, and Manhiça and Bilene-Macia in particular, a strong patriarchal system is in place,

which does not often enable women to make decisions about important affairs of households [34,47], including their health and that of their children. In this region, mothers-in-law also play an important role in children's health and other household matters in the absence of the child father [48]. This gender imbalance may negatively influence the biometric registration of infants and children if the fathers' children and other relevant family members are not involved.

The results of this study show that the study participants used their knowledge to recognise that the biometric registration was a new experience different from that of their past, and they also assessed its potential risks, as Schutz [24] hypothesised. As the results of this study highlight, some caregivers said that other caregivers did not adhere to the biometric registration pilot project because they perceived it as new, strange and different from their common collective experience; despite persistent explanation of the benefits of the registration. Conversely, other participants used their knowledge to evaluate their actual experience of applying the paper-based registration vis-a-vis the potential benefits of the biometric registration, and they welcomed it, notwithstanding the perceived barriers for its use.

Furthermore, all participants of this study also indicated potential opportunity to overcome the perceived barriers and increase the acceptability of the biometric registration. These included the need for greater awareness activities targeting all members of the community, including children's fathers about the usefulness of the biometric registration among infants and children, and the belief of both healthcare providers and caregivers that biometric registration will help solve challenges regarding the identification of the children at the healthcare facility. Healthcare providers, in particular, also mentioned that there were availability of human resources and willingness of implementing the biometric registration; while all caregivers with or without experience of biometric registration perceived adherence to the biometric registration as a healthcare facility norm. This caregivers' perception represents one of the potential facilitators to the biometric registration. However, previous analyses have also shown that the acceptance and adherence to biometric registration can be associated with possible negative consequences for nonparticipation, such as restriction to services [23]. Indeed, most caregivers of this study linked their participation or potential participation to the biometric registration with the potential benefits of their children to healthcare, and they were aware that lack of adherence to the biometric registration could (in the future) potentially prevent their children to access to adequate health services.

The results of this study raise the need for better communication and engagement of the local communities in the biometric registration to prevent perceived coercion and potential exclusion in the participation of the biometric registration. Moreover, all participants, in particular caregivers and their family members, should be offered a good overview of the biometric registration process, how it works and how the collected images will be saved, used and protected.

## Limitations

Given the qualitative nature of this study, its findings are limited to the study setting and the selected participants, and they should not be generalised to other settings. The study also used a prototype biometric system which does not necessarily reflect the performance or usability of the intended final product. The study is also subject to sample-bias because it did not include all potential caregivers with infants and children, and the majority of them were those who accepted to participate in the prototype pilot study. Some views on the prototype itself could not be detached from the quasi-experimental context from which participants were dawn. Moreover, the study did not include all relevant actors, such as children's father,

children's grandmothers and community leaders, who can potentially influence the acceptability of the biometric registration.

## Conclusion

Findings of this study revealed that the piloted biometric prototype registration system is widely accepted among healthcare providers and caregivers. The acceptance of the tested biometric prototype registration is associated with its potential benefits in comparison to the actual registration system–paper-based registration–which is considered inefficient and problematic. All participants of this study used their knowledge and experience to evaluate the potential benefits, facilitators, and barriers to use this system.

Despite this acceptability, future implementations of biometric registration in the community should take into account the perceived perceptions and the social and cultural context of the caregivers, such as gender imbalance within family constructs and potential pre-existing myths and taboos. Moreover, there is a need to promote greater community awareness and engagement involving all relevant stakeholders to maximize the acceptability of any future biometric registration system's implementation.

## Supporting information

**S1 Appendix. Semi-structured interview (SSI) guide for healthcare providers.**
(DOC)

**S2 Appendix. Focus Group Discussion (FGD) guide for caregivers enrolled in the study.**
(DOC)

**S3 Appendix. Focus Group Discussion (FGD) guide for caregivers not enrolled in the study.**
(DOC)

**S4 Appendix. Focus Group Discussion (FGD) guide for data collectors.**
(DOC)

**S5 Appendix. COREQ guideline.**
(PDF)

## Author Contributions

**Conceptualization:** Noni Gachuhi, Rebecca Distler, Quique Bassat, Khátia Munguambe, Charfudin Sacoor.

**Data curation:** Olga Cambaco, Carlos Cuinhane, Estevão Mucavele, Célia Chaúque.

**Formal analysis:** Olga Cambaco, Carlos Cuinhane.

**Funding acquisition:** Charfudin Sacoor.

**Investigation:** Olga Cambaco, Estevão Mucavele, Célia Chaúque.

**Methodology:** Khátia Munguambe.

**Software:** Olga Cambaco, Carlos Cuinhane.

**Supervision:** Khátia Munguambe, Charfudin Sacoor.

**Validation:** Khátia Munguambe, Charfudin Sacoor.

**Writing – original draft:** Olga Cambaco.

**Writing – review & editing:** Olga Cambaco, Noni Gachuhi, Rebecca Distler, Carlos Cuinhane, Emily Parker, Quique Bassat, Franscisco Saute, Khátia Munguambe, Charfudin Sacoor.

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
