## [Decision Letter · Decision Letter 0]

22 Jul 2021

PONE-D-21-18662

Acceptability and perceived facilitators and barriers to the usability of biometric registration among infants and children in Manhiça district, Mozambique: a qualitative study

PLOS ONE

Dear Dr. Cambaco,

Thank you for submitting your manuscript to PLOS ONE. After careful consideration, we feel that it has merit but does not fully meet PLOS ONE’s publication criteria as it currently stands. Therefore, we invite you to submit a revised version of the manuscript that addresses the points raised during the review process.

We look forward to receiving your revised manuscript.

Kind regards,

Livio Provenzi

Academic Editor

PLOS ONE

2. Please revise your supplementary figures as follows: S2 and S3 appear to be the same, and English translations of these documents are needed. Please provide the translations and resolve S2 and S3.

“This study was funded by The Global Good Fund I LLC, (USA) and conducted by Manhiça Foundation/CISM. CISM is supported by the Government of Mozambique and the Spanish Agency for International Development (AECID).”

“We declare no competing interests. The analysis conducted is in absence of commercial or financial gains.”

“We also address our thanks to the data field team (field supervisors, data collectors, transcribers). We also want to acknowledge the project sponsor “The Global Good Fund I LLC, (USA) and our research collaborator ELEMENT Inc.” that have contributed to develop the protocol and data collection tools employed for this study. A big thank you also goes to AECID and ISGlobal.”

Funding information should not appear in the Acknowledgments section or other areas of your manuscript. We will only publish funding information present in the Funding Statement section of the online submission form.

 “This study was funded by The Global Good Fund I LLC, (USA) and conducted by Manhiça Foundation/CISM. CISM is supported by the Government of Mozambique and the Spanish Agency for International Development (AECID).”

Reviewers' comments:

Reviewer's Responses to Questions

**Comments to the Author**

1. Is the manuscript technically sound, and do the data support the conclusions?

Reviewer #1: Partly

Reviewer #2: Yes

2. Has the statistical analysis been performed appropriately and rigorously? 

Reviewer #1: N/A

Reviewer #2: N/A

3. Have the authors made all data underlying the findings in their manuscript fully available?

Reviewer #1: No

Reviewer #2: Yes

4. Is the manuscript presented in an intelligible fashion and written in standard English?

Reviewer #1: Yes

Reviewer #2: Yes

5. Review Comments to the Author

Reviewer #1: The manuscript is a qualitative study and aims at describing how a novel biometric registration prototype of newborns is perceived by health workers, data collectors and parents of infants and children.

The paper is of interest since it addresses a key topic in low-and middle-income countries, where 1 out of 4 newborns is registered at birth. This may cause many risks to children of these countries.

The manuscript well clarifies the problem. However the methodological part is not well described and must be improved. Authors can find below my main comments:

-Introduction lines 111-120: it is not clear if authors are describing the aims of this manuscript (that are stated below at lines 131-135) or those of a previous research (no reference is provided).Please clarify

-Authors often refer to a pilot project that is not described in the paper. I suggest to clarify this point. I suggest to describe the novel biometric registration prototype that uses a non-touch platform using smart phones and tablets to capture physical characteristics of infants and children for electronic registration. In the case this part was already published, I suggest to reference that work.

- I would suggest to explain how the biometric registration was performed and how and where the images were stored and made available to other districts/ healthcare centers. Authors suggest that biometric registration would help in identifying babies if they are stolen but it is not clear how data are available to the community. This is a key point

-I suggest authors to better describe the methodology used to perform focus groups and interviews. Please define who was the moderator of the FGs. How results were evaluated and grouped? How many people assessed focus group results? How did they managed the results? How NVivo software works?

Reviewer #2: The paper presents a qualitative study on acceptability, facilitators and barriers to the adoption of a biometric system for the registration of infants and children in Manhica district, in Mozambique

The paper presents a potentially interesting study, the introduction is clear and the methods section is well presented. There are some spelling errors and typos that should be corrected (for example line 107). Here a few general comments:

Abstract: the introduction is a bit misleading because focused on the description of the biometric system. Please be more focused on the theme of the current work.

There is no agreement within the manuscript on the actual aims of the study: in the Introduction, at LINE 132: The specific focus of this research is on acceptability and perceived facilitators and barriers of using a biometric identity system to register infants and children). However, the methods section states (LINE 212) acceptability, feasibility and usability of the biometric registration among the healthcare providers, and (LINE 217) acceptability, feasibility and usability issues with caregivers and data collectors.

In general, it would be useful to have a more structured discussion according to the findings of the research regarding acceptability, feasibility and usability (if these are the aims).

Methods: this section could be more concise. I appreciate the effort to provide context, but this results in a very long section. It would also be helpful to understand how the guides of the interviews and focus groups were created.

Please find below some specific comments.

Line 77. This sentence seems a repetition of sentence at Line 73. Please check

Lines 111-120.. Providing some insights related to the mentioned study could hep to provide a better perspective of the present study.

Line 118. The aim of the present study appears to be the secondary objective of a larger investigation. However, this is confusing when presented in the manuscript. A paragraph at the end of the introduction section providing clear objectives of the present study would be helpful.

Line 132-135. Same as above, please provide the objective of the study in a single paragraph in the introduction.

Line 189-193. This sentence is unclear, please clarify which are the two communities involved.

Line 210-211. Please provide insights given by the pilot test, or remove the sentence

Table 1: Please provide the number if caregivers participating in the pilot and not.

Line 212-217. Please be clear about the objectives of the semi-structured interviews and the focus groups: were the objectives the same (as it appears to be in the methods section: “ to assess acceptability, usability)

Line 224-225: This is a result, please move to the appropriate section.

Table 2 remove and within bullet points. Please be uniform in the description of the inclusion criteria: the same inclusion criteria is sometimes split into two different bullet points. The same inclusion criteria is sometimes described with different words (Willingness to participate).

Line 279-281. Why is the information about the data collectors not provided in a table, as done with healthcare providers and caregivers?

Table 3. Check number of Male/Female: not in agreement with text

Tables 3-4: Please be consistent with the name of the rows (age groups/age range)

Line 722-724: These appear to be important pilot tests to compare the findings of the present study, could you provide more insights?

Line 792: Do you mean all caregivers of the district? Explain which are all potential caregivers.

6. PLOS authors have the option to publish the peer review history of their article (what does this mean?). If published, this will include your full peer review and any attached files.

Reviewer #1: **Yes: **Emilia Biffi

Reviewer #2: No

---

## [Author Response · Author response to Decision Letter 0]

4 Oct 2021

Subject: Submission of the revised manuscript [PONE-D-21-18662] - [EMID:ce714c6d40f906e4]

Dear editor and reviewers,

Thank you for reviewing our manuscript “Acceptability and perceived facilitators and barriers to the usability of biometric registration among infants and children in Manhiça district, Mozambique: a qualitative study”. The authors of this manuscript have read the current Instructions for Authors, and agreed to accept the recommedned conditions. All authors have also read and agreed upon the submitted version of the manuscript. 

To the Editor:

1. Please ensure that your manuscript meets PLOS ONE's style requirements, including those for file naming. The PLOS ONE style templates can be found at:

Answer: We are thankful for the suggestion. The manuscript has been reviewed and formatted according to the PLOS ONE templates. We believe that the new version comply with the journal requirements.

2. Please revise your supplementary figures as follows: S2 and S3 appear to be the same, and English translations of these documents are needed. Please provide the translations and resolve S2 and S3.

Answer: We agree with the editor. In fact S2 and S3 FGD guides are the same. We have now included the other guide and translated both to English. All the guides are now translated and submitted as supplementary files.

“This study was funded by The Global Good Fund I LLC, (USA) and conducted by Manhiça Foundation/CISM. CISM is supported by the Government of Mozambique and the Spanish Agency for International Development (AECID).”

 Answer: The funding statement was revised and now includes additional information about the role of the funder.

“We declare no competing interests. The analysis conducted is in absence of commercial or financial gains.”

Answer: The competing interest statement included the suggested information and was included in the cover letter according to journal requirements. 

“We also address our thanks to the data field team (field supervisors, data collectors, transcribers). We also want to acknowledge the project sponsor “The Global Good Fund I LLC, (USA) and our research collaborator ELEMENT Inc.” that have contributed to develop the protocol and data collection tools employed for this study. A big thank you also goes to AECID and ISGlobal.”

Funding information should not appear in the Acknowledgments section or other areas of your manuscript. We will only publish funding information present in the Funding Statement section of the online submission form.

 “This study was funded by The Global Good Fund I LLC, (USA) and conducted by Manhiça Foundation/CISM. CISM is supported by the Government of Mozambique and the Spanish Agency for International Development (AECID).”

Answer: All information about funding was removed from the acknowledgment section and included in the cover letter. 

Answer to the Reviewer #1:

Reviewer #1: The manuscript is a qualitative study and aims at describing how a novel biometric registration prototype of newborns is perceived by health workers, data collectors and parents of infants and children.

The paper is of interest since it addresses a key topic in low-and middle-income countries, where 1 out of 4 newborns is registered at birth. This may cause many risks to children of these countries.

The manuscript well clarifies the problem. However the methodological part is not well described and must be improved. Authors can find below my main comments.

-Introduction lines 111-120: it is not clear if authors are describing the aims of this manuscript (that are stated below at lines 131-135) or those of a previous research (no reference is provided). Please, clarify.

Answer: The issues raised were clarified. We have now rephrased and clearly stated the objective of the current study as shown in the line 171-179 of the new version.

-Authors often refer to a pilot project that is not described in the paper. I suggest to clarify this point. I suggest to describe the novel biometric registration prototype that uses a non-touch platform using smart phones and tablets to capture physical characteristics of infants and children for electronic registration. In the case this part was already published, I suggest to reference that work.

- I would suggest to explain how the biometric registration was performed and how and where the images were stored and made available to other districts/ healthcare centres.

Answer: In the new version of the manuscript, we have explained about this project as highlighted in lines 151-172. The implementation process of the pilot project was not published, but we provided a summary of how this project was implements. It is important to add that the project consisted in the testing of a specific device: biometric registration prototype, which was still in the process of its development. This device was used in to take images of the infants and children, and the of objective was to assess the performance and robustness of the device in order to better improve its feasibility and usability in the future biometric registration. Another objective was to evaluate how participants of the experiment perceived, accepted and what barriers could humper the tested device because such an experience was new in the community.

Similarly, we have explained in the lines mentioned how the images were stored. As stated, the images captured were stores in the Institution which carried out the experiment, and only the researcher linked to the development of the device accessed the images. 

Authors suggest that biometric registration would help in identifying babies if they are stolen but it is not clear how data are available to the community. This is a key point.

Answer: We agreed that the information provided can confuse the reader; despite being important to bring this perception, does not specify who suggest. We proceed to clarify the sentence as indicated in lines 672-677 of the new version that this was related to the perceptions of the caregivers and not the personal opinion of the authors..

As it was explained earlier, the pilot project did not include availability of the images to the communities and health facilities. The images were only captured, stored and used for experimental purpose. All participants involved in the experiment were explained and informed about the future potential usefulness of the piloted prototype if it were definitely implemented in the community. Moreover, they were aware that those images were for experimental of the device in the development, which possible could be used in the community in the future. Therefore, participants of this qualitive study, in turn, referred to the potential usefulness of the biometric prototype in the communities.

-I suggest authors to better describe the methodology used to perform focus groups and interviews. Please define who was the moderator of the FGs. How results were evaluated and grouped? How many people assessed focus group results?

Answer: The new version of the manuscript presents the required explanation in lines 396-403 (procedure section) as well as additional information in data analysis section, lines 406-418. We provided a summary of the procedures, but no names were indicated as the qualitative methodology manuals recommend. Semi-structured interviews and focus group discussions are data collection tools well described in different manuals of methodology (just to mention some methodological sources: Creswell, 2014; Richardson, 2008; Bernard, 2006; Quivy & Campenhoudt, 2008; Denzin & Lincoln (1994). We followed all procedures recommended in these manuals, and we hope the summary provided will help to clarify the above questions.

How did they managed the results? How NVivo software works? 

Answer: The section of data analysis clearly sated how the data were managed. Additional information was proved in lines 406-423. Nvivo Software is a qualitative package for data analysis. It is similar to SPSS package which is used to analyse quantitative data. To use a Nvivo, one has to first read a sample of transcriptions, second, cod the data and stablish the categories and its properties, third, insert the categories and subcategories (if available) and the transcriptions to the software; fourth, read and codding each transcription from the beginning to the end according to the issues of each category or subcategory. After all transcriptions were codded, a summary of the information is then displayed, which can be analysed according to the emerging themes. More information about Nvivo can be obtained in the Nvivo guideline in the following link: www.qsrinternational.com

Answer to Reviewer #2:

Reviewer #2: The paper presents a qualitative study on acceptability, facilitators and barriers to the adoption of a biometric system for the registration of infants and children in Manhica district, in Mozambique

The paper presents a potentially interesting study, the introduction is clear and the methods section is well presented. There are some spelling errors and typos that should be corrected (for example line 107). Here a few general comments:

Answer: Thank you for the revision of the manuscripts and the details presented. We have addressed all suggestions and comments as presented in issue presented below.

Abstract: the introduction is a bit misleading because focused on the description of the biometric system. Please be more focused on the theme of the current work.

Answer: The abstract was rephrased and the new version focuses on the theme of the study object. However, a general context was maintained as a way of informing the reader why the current theme was relevant. Moreover, the Journal recommend a contextualization of the study object.

There is no agreement within the manuscript on the actual aims of the study: in the Introduction, at LINE 132: The specific focus of this research is on acceptability and perceived facilitators and barriers of using a biometric identity system to register infants and children). However, the methods section states (LINE 212) acceptability, feasibility and usability of the biometric registration among the healthcare providers, and (LINE 217) acceptability, feasibility and usability issues with caregivers and data collectors.

Answer: Indeed, the draft submitted had had disagreement in terms of the objectives as the preliminary objectives were those of the piloted objectives. We have addressed this issue, and we have harmonized the objectives of the current qualitative study as shown in lines: 174-179. Now, these objectives are in agreements with those of FGDs and interviews guides in lines: 356-359.

In general, it would be useful to have a more structured discussion according to the findings of the research regarding acceptability, feasibility and usability (if these are the aims).

Answer: Yes, we agree with this suggestion. The actual discussion presented in the new version focus only on the results of this research findings.

Methods: this section could be more concise. I appreciate the effort to provide context, but this results in a very long section. It would also be helpful to understand how the guides of the interviews and focus groups were created.

Answer: We acknowledge that this is a relevant issue and understand that there is need to clarified how guide used in the study were created. The interviews and guides of the FGDs were designed according to the study goals and target groups. First, FGD with the mothers recruited in the study and data collector guides were designed focusing the relevant elements of the piloted project, such perceptions of the participants who had participated in the project about acceptability and barriers. Second, the FGD with caregivers who had not participated in the piloted project was created to assess how participants of other communities that had not involved in the piloted project would accept the biometric prototype implemented in their communities, what could hinder its implementation. This was relevant to generate information which could inform better the future implementation of the biometric prototype. All guides were discussed by all members of the research team before the commencement of the study. We have provided a summary of how the guides were designed in the new manuscript version as shown in lines: 360-386.

Please find below some specific comments.

Line 77. This sentence seems a repetition of sentence at Line 73. Please check

Answer: We have checked and rephrased the sentence, as presented in lines: 109-114 of the new manuscript version.

Lines 111-120.. Providing some insights related to the mentioned study could hep to provide a better perspective of the present study.

Answer: The insights related to the mentioned study was presented in lines 151-172 of the manuscript version. More detail was provide in answer number 2 of the Reviewer # 1 above.

Line 118. The aim of the present study appears to be the secondary objective of a larger investigation. However, this is confusing when presented in the manuscript. A paragraph at the end of the introduction section providing clear objectives of the present study would be helpful.

Line 132-135. Same as above, please provide the objective of the study in a single paragraph in the introduction.

Answer: The objectives of the current study were reformulated and clearly stated in the introduction, lines: 174-179.

Line 189-193. This sentence is unclear, please clarify which are the two communities involved. 

Answer: The sentence was rephrased and the communities involved were clearly indicated as stated in lines: 268-282 of the new manuscript version.

Line 210-211. Please provide insights given by the pilot test, or remove the sentence

Answer: As the results were of pilot test were not different, we removed the sentence.

Table 1: Please provide the number if caregivers participating in the pilot and not. to add a line in the table showing caregiver who did not participate in the pilot study.

Answer: Table 1 was redrawn, and the new one present clear difference between participants who participated in the pilot and those who did not.

Line 212-217. Please be clear about the objectives of the semi-structured interviews and the focus groups: were the objectives the same (as it appears to be in the methods section: “ to assess acceptability, usability)

Answer: We are thankful for the suggestion. Indeed, for each target group different objectives were defined and were now clarified. The objective of the semi-structured interviews with the health care providers was: to describe the perceptions of health care providers regarding the feasibility, usability and acceptability of the biometric device in infants and child biometric prototype.

The FGDs with caregivers was: to determine the perceptions of caregivers of subjects regarding the usability of infant and child biometric prototype as well as perceived barriers.

The FGDs with data collectors was: to identify experiences, barriers and facilitators to the acceptability and use of the infant and child biometric device among data collectors.

The idea was to collect similar information in different groups for data triangulation.

Line 224-225: This is a result, please move to the appropriate section.

Answer: We appreciate this suggestion, but we think that inclusion criteria are part of the methodological section as recommended in several qualitative manual books. This highlight who was selected to the interview or FGDs.

Table 2 remove and within bullet points. Please be uniform in the description of the inclusion criteria: the same inclusion criteria is sometimes split into two different bullet points. The same inclusion criteria is sometimes described with different words (Willingness to participate). 

Answer: The Table 2 was corrected, and it is now according to the recommendations.

Line 279-281. Why is the information about the data collectors not provided in a table, as done with healthcare providers and caregivers? 

Answer: Thank you for highlighting this point. We have revised and included the table with demographic information of the data collectors. It was now stated as Table 5.

Table 3. Check number of Male/Female: not in agreement with text

Answer: The Table 3 is now correct.

Tables 3-4: Please be consistent with the name of the rows (age groups/age range):

Answer: The Tables 3 and 4 have been harmonized.

Line 722-724: These appear to be important pilot tests to compare the findings of the present study, could you provide more insights? 

Answer: more information about the findings of the piloted test was provided in lines: 938-943

Line 792: Do you mean all caregivers of the district? Explain which are all potential caregivers. 

Answer: the sentence was rephrased and it is now clear that the referred caregivers were those from the Manhiça district, as shown in lines 975-976.

References

Bernard, H. R. (2006). Research methods in Anthropology. Qualitative and quantitative approaches, 4th edn. New York: Altamira Press.

Creswell, J.W. (2014). Research Design. Qualitative, quantitative and mixed methods approaches. California: Sage Publications. Cap.

Denzin, N.K. e Lincoln, Y.S. (1994). Handbook of Qualitative Research. London: Sage Publications. Cap. 13 e 17.

Quivy, R. & Campenhoudt, L. V. 2008. Manual de Investigação em Ciências Sociais. Gradiva, Lisboa. 

Richardson, R.J. 2008. Pesquisa Social. Métodos e Técnicas. Atlas, São Paulo.

---

## [Decision Letter · Decision Letter 1]

15 Nov 2021

Acceptability and perceived facilitators and barriers to the usability of biometric registration among infants and children in Manhiça district, Mozambique: a qualitative study

PONE-D-21-18662R1

Dear Dr. Cambaco,

We’re pleased to inform you that your manuscript has been judged scientifically suitable for publication and will be formally accepted for publication once it meets all outstanding technical requirements.

Kind regards,

Livio Provenzi

Academic Editor

PLOS ONE

Additional Editor Comments (optional):

Reviewers' comments:

Reviewer's Responses to Questions

**Comments to the Author**

1. If the authors have adequately addressed your comments raised in a previous round of review and you feel that this manuscript is now acceptable for publication, you may indicate that here to bypass the “Comments to the Author” section, enter your conflict of interest statement in the “Confidential to Editor” section, and submit your "Accept" recommendation.

Reviewer #1: (No Response)

Reviewer #2: All comments have been addressed

2. Is the manuscript technically sound, and do the data support the conclusions?

Reviewer #1: Partly

Reviewer #2: Yes

3. Has the statistical analysis been performed appropriately and rigorously? 

Reviewer #1: N/A

Reviewer #2: N/A

4. Have the authors made all data underlying the findings in their manuscript fully available?

Reviewer #1: No

Reviewer #2: Yes

5. Is the manuscript presented in an intelligible fashion and written in standard English?

Reviewer #1: No

Reviewer #2: Yes

6. Review Comments to the Author

Reviewer #1: (No Response)

Reviewer #2: The authors have addressed all comments and no further action is needed. The manuscript can be published in the present form.

7. PLOS authors have the option to publish the peer review history of their article (what does this mean?). If published, this will include your full peer review and any attached files.

Reviewer #1: No

Reviewer #2: No

---

## [Editor Report · Acceptance letter]

9 Dec 2021

PONE-D-21-18662R1 

Acceptability and perceived facilitators and barriers to the usability of biometric registration among infants and children in Manhiça district, Mozambique: a qualitative study 

Dear Dr. Cambaco:

I'm pleased to inform you that your manuscript has been deemed suitable for publication in PLOS ONE. Congratulations! Your manuscript is now with our production department. 

Kind regards, 

on behalf of

Dr. Livio Provenzi 

Academic Editor

PLOS ONE